Unlocking the potential of ancient hexaploid Indian dwarf wheat, Tritium sphaerococcum for grain quality improvement

Adhikari Sneha 1
Kumari Jyoti 2 jj.gene@gmail.com
Bhardwaj Rakesh 2
Jacob Sherry 2
Langyan Sapna 2
Sharma Shivani 2
M. Singh Anju 3
Kumar Ashok 2
1 ICAR-Indian Institute of Wheat and Barley Research , Shimla, HP , India
2 ICAR-National Bureau of Plant Genetic Resources , New Delhi, Delhi , India
3 ICAR-Indian Agricultural Research Institute , New Delhi, Delhi , India
Lazo Gerard
Electronic publication date: 2023 Jul 27
Publication date: 2023
Volume: 11
Electronic Location ID: e15334
Received 2022 Sep 16; Accepted 2023 Apr 11
Copyright: © 2023 Adhikari et al.
Copyright year: 2023
Copyright holder: Adhikari et al.
License: This is an open access article distributed under the terms of the Creative Commons Attribution License, which permits unrestricted use, distribution, reproduction and adaptation in any medium and for any purpose provided that it is properly attributed. For attribution, the original author(s), title, publication source (PeerJ) and either DOI or URL of the article must be cited.
License URL: https://creativecommons.org/licenses/by/4.0/

Keywords: Triticum sphaerococcum, Grain quality, Diversity, Evaluation, Indian dwarf wheat

Funding: NBPGR In-House Project PGR/DGE-BUR-DEL-01.01 The research work was carried out under NBPGR In-House Project (PGR/DGE-BUR-DEL-01.01). The funders had no role in study design, data collection and analysis, decision to publish, or preparation of the manuscript.

==============================
Wild and ancient wheat are considered to be a rich source of nutrients and better stress tolerant, hence being re-considered for mainstreaming its cultivation by the farmers and bringing it back to the food basket. In the present study, thirty-four diverse accessions of Indian dwarf wheat, Triticum sphaerococcum conserved in the Indian National Genebank were evaluated for thirteen-grain quality parameters namely thousand-grain weight (TGW), hectolitre weight (HW), sedimentation value (Sed), grain hardness index (HI), protein (Pro), albumin (Alb), globulin (Glo), gliadin (Gli), glutenin (Glu), gluten, lysine (Lys), Fe2+ and Zn2+ content, and four antioxidant enzymes activities. Substantial variations were recorded for studied traits. TGW, HW, Sed, HI, Pro, Alb, Glo, Gli, Glu, Gluten, Lys, Fe2+, and Zn2+ varied from 26.50–45.55 g, 70.50–86.00 kg/hl, 24.00–38.00 ml, 40.49–104.90, 15.34–19.35%, 17.60–40.31 mg/g, 10.75–16.56 mg/g, 26.35–44.94 mg/g, 24.47–39.56 mg/g, 55.33–75.06 mg/g, 0.04–0.29%, 42.72–90.72 ppm, and 11.45–25.70 ppm, respectively. Among antioxidants, peroxidase (POX), catalase (CAT), glutathione reductase (GR), and superoxide dismutase (SOD) activity ranged from 0.06–0.60 unit/ml, 0.02–0.61 unit/ml, 0.11–2.26 unit/ml, and 0.14–0.97 unit/ml, respectively. Hardness Index was positively associated with Pro and Zn2+ content whereas Lys was negatively associated with gluten content. Likewise, gluten and Fe2+ content had a positive association with the major protein fraction i.e., Gli and Glu. Hierarchical cluster analysis grouped 34 accessions into four clusters and the major group had nine indigenous and eight exotic accessions. We also validated high GPC accessions and EC182958 (17.16%), EC187176 and EC182945 (16.16%), EC613057 (15.79%), IC634028 (15.72%) and IC533826 (15.01%) were confirmed with more than 15% GPC. Also, superior trait-specific accessions namely, EC187167, IC534021, EC613055, EC180066, and EC182959 for low gluten content and IC384530, EC313761, EC180063, IC397363, EC10494 for high iron content (>76.51) were identified that may be used in wheat quality improvement for nutritional security of mankind.

Introduction

Worldwide, wheat is the most important cereal crop, contributing 55% of total carbohydrates, 21% of the total proteins and 19-20% of the calories demand of around half of the world’s population (Shiferaw et al., 2013). Along with carbohydrates, proteins, dietary fiber, and micronutrients, wheat also contains minerals, vitamins, and fats (Sarwar et al., 2013). Grain quality consists of various characteristics that together are responsible for determining the end use of grains (Rondanini, Borrás & Savin, 2013). The chemical composition of grain i.e., starch content is considered important for ethanol production whereas protein and carbohydrate content for food and feed production. The major component of wheat endosperm is starch and it also consists of glucose polymers, amylase, amylopectin, and a small amount of protein and lipids (Fasahat, Rahman & Ratnam, 2015). The wheat grains milled into flour is used as raw material for different foods such as chapatti, bread, cakes, biscuits, muffins, pasta, noodles, sweets, and pastries (Uthayakumaran & Wrigley, 2017). The utilization of wheat grain in different forms mainly depends on protein content and gluten strength.

In bread wheat (Triticum aestivum), protein content varies from ~9–13% (Žilić et al., 2011; Mohan & Gupta, 2013; Punia, Sandhu & Siroha, 2019). Wheat protein is comprised of four fractions (Osborne, 1907) which include albumins (water soluble), globulins (salt soluble), gliadins (alcohol soluble), and glutenins (diluted acid or base soluble). The 25% fraction of total wheat protein is contributed by albumin and globulin components (Belderok et al., 2000). The remaining two protein fractions, gliadins, and glutenins are collectively called gluten and share 75% of total protein and provide visco-elastic properties to the wheat dough. Therefore, the proportions of these four components are responsible for dough making and baking quality of flour (Vigni et al., 2013). Sedimentation value provides an idea about the content and quality of gluten as well as loaf volume (Pasha et al., 2007) and thus is helpful in the classification of wheat in different categories for their utilization. Grain hardness is influenced by protein content and moisture content (Turnbull & Rahman, 2002) and causes notable effects on milling as well as baking quality (Bettge, Morris & Greenblatt, 1995). Many critical factors influence the quality of wheat end-product hence it is important to assess parameters for global trade. Based on grain hardness, wheat is grouped into two categories i.e., soft wheat and hard wheat (Pomeranz & Williams, 1990).

Free radicals and reactive oxygen species (ROS) cause harmful effects on the cell membrane, enzymes, DNA, and maintenance of DNA. Oxidative damage to DNA and enzymes can result in some serious health issues such as cardiovascular diseases, carcinogenesis, type II diabetes, and obesity. Free radicals have been linked to over 60 kinds of diseases. This includes cancer, Alzheimer’s disease, Parkinson’s disease, and cataracts. Enzymatic antioxidants such as glutathione reductase (GR), superoxide dismutase (SOD), catalase (CAT), and peroxidase (POX) (Narwal et al., 2014) directly react with ROS (Lobo et al., 2010) and consequently prevents oxidative damage resulting in reduced risk of oxidative-stress related chronic diseases. Enzymatic antioxidants help in the breakdown of free radicals by seeking out oxidative products and converting them into hydrogen peroxide, and then into the water. In addition, they also act as metal chelators and terminate the oxidative enzyme inhibitors hence ROS reactions (Karadag, Ozcelik & Saner, 2009). Hence the improved concentration of antioxidants helps in disease prevention & health promotion. Further, regarding mineral content, cereals and their products contribute approximately 50% of Fe2+ and 20% of Zn2+ in the Western diet (Velu et al., 2017). In case of wheat, Fe2+ and Zn2+ content was reported to be low and that too was situated in the outer layers of grains. Presently around three billion people are malnourished due to the non-availability of micronutrients as well as vitamins. A less diverse diet leads to deficiency of minerals, among which Fe2+ and Zn2+ deficiency causes more pronounced effects due to involvement in various metabolic processes. Micronutrient enrichment of major food crops through breeding is one of the sustainable ways to reduce malnutrition among higher-risk groups (resource-poor women, infants, and children) (Šramková, Gregová & Šturdík, 2009). It has been proven by various researchers that in the wheat genome traits related to micro-nutrient enrichment are available that could facilitate significant improvement in vitamins and minerals as well as growth-promoting components without any adverse effect on yield (Rose et al., 2001; Šramková, Gregová & Šturdík, 2009). During the course of the domestication and selection process, desirable allelic combinations were selected which led to the evolution of modern crops that feed the world today but at the cost of reduced variation resulting in the narrowing down of the genetic base of crop species (Van Heerwaarden et al., 2011) and is referred as domestication bottleneck (Hammer, 1984). For the development of high-yielding wheat varieties with improved quality for targeting specific products, several breeding programs have been initiated. As a consequence of this, wheat cultivation is based on genetically similar cultivars which resulted in a compromised genetic base due to dependency on the limited number of parents (Shuaib et al., 2010).

Wheat-related species are considered potential sources for grain yield and quality traits for which these are not fully explored. Therefore, an urgent need was sought for the characterization of these species with the aim of identification of promising lines for improved quality traits. Among ancient wheat, Triticum sphaerococcum (AABBDD, 2n = 6x = 42) is a landrace of wheat and is known as the Indian dwarf wheat. It has several advantageous features such as short and strong culms, hemispherical grains, higher protein content compared to bread wheat, and resistance to biotic and abiotic stresses, however, sphaerococcum wheat is poorly studied (Josekutty, 2008, Matsuoka, 2011). Being a hexaploid species, it may be easily exploited for bread wheat improvement to achieve food and nutritional security to fulfill United Nations’ sustainable development goal. Therefore, the main objectives of this study were (1) to evaluate 34 accessions of T. sphaerococcum for important quality traits and identification of superior line for individual traits and (2) diversity assessment of T. sphaerococcum accessions based on quality traits.

Materials and Methods

The experimental material for the present investigation comprised a diverse set of thirty-four accessions of T. sphaerococcum conserved in the national genebank namely IC384530, IC397363, IC533826, IC53387, IC534021, IC534522, IC212160, IC534882, IC420038, EC313761, EC541164, EC576654, EC613055, EC613057, EC187172, EC187182, EC187181, EC180062, EC187183, EC187167, EC182959, EC182945, EC10494, EC10492, EC182958, EC180041, EC180061, EC180067, EC182947, EC180066, EC182956, EC180063, EC187176, IC634028. The grain harvest was obtained from the field trial grown during Rabi 2019-20 at ICAR-National Bureau of Plant Genetic Resources (NBPGR), Issapur farm (7650′E longitude and 2840′N latitude, 223 m MSL). The soil of the farm was sandy loam in texture having organic carbon 0.7%, available N 297.6 kg/ha, available P 26.7 kg/ha, and available K 196.0 kg/ha with soil pH of 6.5. Recommended agronomic practices, irrigation, and fertilizers (120 kg N; 60 kg P2O5 and 40 kg K2O ha−1) were followed throughout the experiments. Complete doses of K2O and P2O5 were applied at the time of sowing. Nitrogen was supplied in three split doses, 60 kg N ha−1 at sowing, 30 kg N ha−1 at crown root initiation, and 30 kg N ha−1 at booting stages. The experiment was adequately irrigated with five irrigations during the crop growth. Weeding was done manually. The spray of fungicide, tilt was done to avoid the infection of rust diseases.

These accessions were evaluated biochemically following a completely randomized design with three replications for seventeen quality parameters namely thousand-grain weight (TGW), hectolitre weight (HW), sedimentation value (Sed), grain hardness index (HI), protein content (Pro), albumin content (Alb), globulin content (Glo), gliadin content (Gli), glutenin content (Glu), gluten content, lysine content (Lys), Fe2+ and Zn2+ content, peroxidase activity (POX), catalase activity (CAT), glutathione reductase activity (GR) and superoxide dismutase activity (SOD). The grain material was sun-dried. Grain moisture content and TGW were recorded before the biochemical estimation. For HW and HI whole grain was analyzed whereas, for the rest of the traits, flours were utilized. The flour of seeds of each accession was prepared by grinding these seeds in a cyclotec laboratory flour mill and was tested in three replicates. HW was estimated by using hectolitre apparatus. The sodium dodecyl sulphate (SDS Solution) sedimentation volume of each sample was estimated by the method suggested by Axford, McDermott & Redman (1978). HI and moisture content was analyzed using the single-kernel characterization system (SKCS) model 4100 (Perten Instruments North America, Inc., Reno, NV). Lysine was calculated by the dye-binding method as suggested by Udy (1956). Total protein content was estimated on a wet basis following Micro-Kjeldahl Method (Kjeldahl, 1883). Following the sequential Osborne extraction procedure (Osborne, 1907) the proportion of Alb, Glo, Gli, and Glu was extracted as described by Mughal et al. (2020). As gluten is composed of Gli and Glu therefore it was derived by summing up of estimation of these two contributing components. Fe2+ and Zn2+ content was estimated by wet digestion (15 ml acid mixture (nine concentrated nitric acid: two sulfuric acid: one perchloric acid + 0.5 g wheat flour) followed by atomic absorption spectroscopy (Jackson, 1973). Quantification of extracted proteins (Pro, Alb, Glo, Gli, and Glu) was conducted using a spectrophotometer, and BSA (Bovine Serum Albumin) was used as a standard. The enzymatic antioxidants were estimated by using the methodology suggested by Todorova et al. (2021). Eight promising accessions for grain protein content were validated by growing consecutively at NBPGR Pusa Farm and NBPGR, Issapur Farm, New Delhi during the year 2019–2020, 2020–2021, and 2021–2022.

Statistical analysis

All data were subjected to analysis of variance (ANOVA) using completely randomized design (CRD) and a comparison of treatment means was performed using Duncan’s multiple range test (DMRT) at P < 0.05 using IBM SPSS Statistics software. The mean values of individual genotypes for each trait were subsequently used for analyzing summary statistics and graphs using MS Excel software. Histogram and correlation among traits were analyzed by using IBM SPSS Statistics software. Hierarchical cluster analysis was performed using Euclidean distance measure following Unweighted Pair Group Method with Arithmetic Averages (UPGMA) method by using PAleontological STatistics (PAST) software (Hammer, Harper & Ryan, 2001). The flow of work done is depicted in Fig. S1.

Results

Variation for quality characteristics among T. sphaerococcum accessions

Analysis of variance revealed significant variance for all the characters at 1% level of significance (Table 1). Further to find out the significant differences among genotypes, a post hoc test, Duncan’s multiple range test (DMRT) was performed which resulted in the grouping of accessions in a larger number of subgroups: eight (TGW), 17 (HW), 13 (Sed), 18 (HI), 13 (Pro), 14 (Alb), 14 (Glo), 20 (Gli), 18 (Glu), 19 (Gluten), 19 (Lys), 18 (Fe2+), 13 (Zn2+), 19 (PER), 16 (CAT), 25 (GR), 13 (SOD), which indicates the existence of sufficient variation. Likewise, the existence of wider variation was also supported by other parameters (Table 2). It was also depicted pictorially by the histogram (Fig. S2) and bar diagram (Figs. 1–4). All the traits followed normal distribution except Alb, CAT, and SOD which were positively skewed.

Table 1 Analysis of variance (ANOVA) for different quality characters in T. sphaerococcum accessions.

Source	d. f.	Mean sum of squares	
TGW	HW	Sed	HI	Pro	Alb	Glo	Gli	Glu	Gluten	Lys	Fe+2	Zn+2	PER	CAT	GR	SOD	
Treatment	33	51.572**	33.693**	35.684**	774.831**	4.080**	53.221**	8.061**	47.596**	34.724**	71.924**	0.015**	906.097**	36.213**	0.049**	0.038**	0.971**	0.077**	
Error	68	13.109	0.250	0.250	0.757	0.342	0.469	0.200	0.587	0.594	2.375	0.001	19.974	6.970	0.001	0.001	0.001	0.001	
Note:

TGW, Thousand grain weight; HW, Hectolitre weight; Sed, Sedimentation value; HI, Hardness index; Pro, Protein content; Alb, Albumin content; Glo, Globulin content; Gli, Gliadin content; Glu, Glutenin content; Lys, Lysine content; Fe+2, Iron Content; Zn+2, Zinc content; POX, Peroxidase; CAT, Catalase, GR, Glutathione reductase; SOD, Superoxide dismutase. **1% level of significance.

Table 2 Mean performance and variability for grain quality traits among 34 accessions of T. sphaerococcum.

S.No.	Quality trait	Mean	Range	SE	CV (%)	SED	Top five accessions along with DUNCAN group	
1	TGW	32.85	26.50–45.55	0.71	12.61	0.23	IC420038A
IC534021B
EC187167B
EC180067B,C
EC613057B,C,D
(>36.62)	
2	HW (kg/hl)	81.25	70.50–86.00	0.57	4.12	0.25	EC180066A
EC541164B
IC534522B
IC212160B,C
EC613057 (>84)	
3	Sed (ml)	30.38	24.00–38.00	0.59	11.35	0.24	EC180063A
EC182958B
EC10492B,C
IC534021C
EC187176C (>33)	
4	HI	82.14	40.49–104.91	2.76	19.57	0.75	EC182959A
IC533826B
IC634028B,C
EC182956C,D
EC313761D,E (>94)	
5	Pro (%)	17.62	15.34–19.35	0.20	6.62	0.34	EC613057A
EC182958B,C
EC187176B,C,D
IC634028B,C,D
EC182945B,C,D
(>19.70)	
6	Alb (mg/g)	23.91	17.60–40.31	0.72	17.62	0.46	IC534522A
EC10492B
EC10494B
IC53387C
EC187182C,D (>27.80)	
7	Glo (mg/g)	13.35	10.75–16.56	0.28	12.28	0.20	EC182945A
EC180062B,C
EC180063B,C
EC187172B,C
IC534021C,D (>15.62)	
8	Gli (mg/g)	33.97	26.35–44.94	0.68	11.73	0.58	EC313761A
IC384530B
EC187182C
IC533826C,D
IC397363C,D (>38.42)	
9	Glu (mg/g)	31.00	24.47–39.56	0.58	10.97	0.59	EC182958A
EC10494B
EC180067B,C
EC10492C,D
IC634028D,E (34.64)	
10	Gluten (mg/g)	64.97	55.33–75.06	0.84	7.54	2.37	EC313761A
EC10492A,B
IC384530A,B,C
IC533826B,C,D
EC182958B,C,D (>71.44)	
11	Lys (% of sample)	00.15	0.04–0.29	0.01	47.90	0.001	EC10492A
EC187176A
EC182945B
IC534522C
EC313761C,D (>0.24)	
12	Fe2+Content (ppm)	65.11	42.72–90.72	1.70	15.25	6.97	IC384530A
EC313761B
EC180063B,C
IC397363B,C
EC10494B,C (>76.51)	
13	Zn2+ content (ppm)	19.00	11.45–25.70	0.60	18.29	3.10	EC10492A
EC187182A,B
EC182958A,B,C
EC180041A,B,C
EC313761A,B,C (>23.35)	
14	POX (unit/ml)	0.22	0.06–0.60	0.02	30.46	0.001	EC541164A
IC534882B
EC187167C
EC313761C,D
EC576654D,E (>0.34)	
15	CAT (unit/ml)	0.12	0.02–0.61	0.02	42.32	0.002	EC182945A
EC613057B
EC180061C
IC53387D
EC180041E (>0.20)	
16	GR (unit/ml)	0.70	0.11–2.26	0.10	18.56	0.001	EC180063A
EC180066B
EC182956C
EC10494D
EC187182E (>1.45)	
17	SOD (unit/ml)	0.31	0.14–0.97	0.03	26.37	0.001	EC182956A
IC384530B
EC180062C
EC180067C
EC182947C (>0.48)	
Note:

TGW, Thousand grain weight; HW, Hectolitre weight; Sed, Sedimentation value; HI, Hardness index; Pro, Protein content; Alb, Albumin content; Glo, Globulin content; Gli, Gliadin content; Glu, Glutenin content; Lys, Lysine content; Fe+2, Iron Content; Zn+2, Zinc content; POX, Peroxidase; CAT, Catalase; GR, Glutathione reductase; SOD, Superoxide dismutase; SE, Standard Error; CV, Coefficient of variation.

Figure 1 Variability of various grain quality traits among 34 accessions of T. sphaerococcum for hectolitre weight, sedimentation value and hardness index.

Figure 2 Variability of various grain quality traits among 34 accessions of T. sphaerococcum for protein, gluten and lysine content.

Figure 3 Variability of various grain quality traits among 34 accessions of T. sphaerococcum for iron and zinc content.

Figure 4 Proportion of four protein fractions among 34 accessions of T. sphaerococcum.

Hectolitre weight (HW), SDS sedimentation (Sed) test and grain hardness (HI)

Hectolitre weight (HW) ranged from 70.50 kg/hl (EC187182) to 86.00 kg/hl (EC180066) with the mean of 81.25 kg/hl among studied T. sphaerococcum accessions (Fig. S2A). High HW along with high TKW (26.05–45.55 g) (Fig. S2B) indicated high milling yield potential of Indian dwarf wheat. Gluten quality which is estimated in terms of Sed varied from 24.00 ml (EC182945) to 38.00 ml (EC180063) with an average of 30.38 ml (Figs. S2C and 1). Based on Sed, wheat can be classified into three categories (<48 ml (weak), 48–68 ml (medium strong), >68 ml (strong)) but all 34 accessions of T. sphaerococcum showed Sed less than 48.00 ml therefore grouped in weak gluten class due to its low hydration capacity. Further, they have been classified into three categories based on their suitability for making different products (Table 3). For making good quality bread, chapatti, and biscuit, required Sed are >60 ml, 30–60 ml, and <30 ml, respectively. Further, non-significant but positive estimates of correlation coefficient (+0.18) were found between Sed and protein (Table 4).

Table 3 Classification of T. sphaerococcum accessions based on their sedimentation value and suitability for products.

Sedimentation value (cc)	Suitable for	T. sphaerococcum accessions	
>60 ml	Bread	-	
30–60 ml	Chapatti	IC384530, IC397363, IC533826, IC53387, IC534021, IC534522, IC212160, IC534882, IC420038, EC10494, EC313761, EC541164, EC576654, EC613055, EC613057, EC187172, EC187182, EC187181, EC180062, EC187183, EC187167, EC182959, EC182945, EC10492, EC182958, EC180041, EC180061, EC180067	
<30 ml	Biscuit	EC182947, EC180066, EC182956, EC180063, EC187176, IC634028	

Table 4 Pearson correlation matrix among seventeen grain quality traits including TGW.

	HW	Sed	HI	Pro	Alb	Glo	Gli	Glu	Gluten	Lys	Fe+2	Zn+2	TGW	POX	CAT	GR	SOD	
HW	1																	
Sed	−0.18	1.00																
HI	0.19	−0.23	1.00															
Pro	0.06	0.18	0.37*	1.00														
Alb	−0.03	0.01	0.07	−0.27	1.00													
Glo	−0.16	0.00	−0.12	0.16	−0.16	1.00												
Gli	−0.06	−0.05	−0.12	0.04	0.17	0.25	1.00											
Glu	0.11	0.00	−0.30	0.15	−0.03	−0.08	−0.13	1.00										
Gluten	0.03	−0.04	−0.31	0.14	0.12	0.15	0.73**	0.60**	1.00									
Lys	0.12	0.02	−0.20	−0.03	0.15	0.03	0.25	0.21	−0.35*	1.00								
Fe+2	−0.13	0.17	−0.23	0.27	0.08	0.13	0.48**	0.18	0.52**	0.28	1.00							
Zn+2	−0.22	0.10	0.55**	0.05	0.25	0.07	0.19	0.22	0.31	0.05	0.43*	1.00						
TGW	0.00	0.19	−0.35*	0.06	0.17	0.31	−0.03	0.03	0.00	0.06	0.06	0.22	1.00					
POX	0.00	−0.30	0.40*	−0.38*	0.14	−0.11	−0.02	−0.21	−0.16	−0.25	−0.26	−0.12	−0.29	1.00				
CAT	0.18	-0.34*	−0.20	0.23	−0.33*	0.22	0.06	−0.07	0.00	0.08	0.01	0.04	−0.03	−0.28	1.00			
GR	−0.37*	0.01	0.05	0.25	0.11	0.25	−0.02	−0.13	−0.10	0.03	−0.09	−0.22	0.03	−0.12	−0.27	1.00		
SOD	−0.17	−0.12	0.23	0.17	−0.32	0.21	−0.04	0.06	0.01	−0.24	0.05	-0.33*	0.03	0.20	−0.13	0.23	1.00	
Note:

TGW, Thousand grain weight; HW, Hectolitre weight; Sed, Sedimentation value; HI, Hardness index; Pro, Protein content; Alb, Albumin content; Glo, Globulin content; Gli, Gliadin content; Glu, Glutenin content; Lys, Lysine content; Fe+2, Iron Content; Zn+2, Zinc content; POX, Peroxidase; CAT, Catalase; GR, Glutathione reductase; SOD, Superoxide dismutase. **1% level of significance, *5% level of significance.

Grain hardness ranged from 40.49 (EC182958) to 104.92 (EC182959) with an average of 82.03 (Figs. S2D and 1). Based on HI, accessions were classified in four categories i.e., <33 (Soft), 34–46 (Medium soft), 47–59 (Medium-hard), and >60 (Hard) according to Morris et al. (2001). The majority of accessions (82.35%) were categorized as hard wheat and five accessions EC187176, EC187182, EC182945, EC10492, and EC10494 were grouped in the medium hard category. Only one accession, EC182958, was reported medium soft with HI of 40.49. In our experiment, we observed a significant positive correlation (r = +0.37*) between Pro and HI, however, HI and TGW were negatively associated (−0.35*) (Table 4).

Total protein content (Pro), protein fraction and lysine content (Lys) estimation

Accessions significantly differed from each other for protein content and ranged from 15.34–19.35% with an average of 17.62% (Figs. S2E and 2). Maximum protein content, 19.35% was reported in accession, EC613057 followed by 19.24% in EC10494. The proportion of four protein fractions among 34 accessions of T. sphaerococcum is shown in Fig. 2. The concentration of Alb varied from 17.60 mg/g to 40.31 mg/g with the mean of 23.91 mg/g (Fig. S2F). Only one, IC534522, exhibited a moderate concentration of Alb (40.32 mg/g) whereas rest consisted of low concentration of Alb ranging from 17.60 mg/g to 30.00 mg/g (Table 5). As far as Glo is concerned, 18 accessions showed moderate concentration and it ranged from 14.00–16.56 mg/g with the highest concentration of Glo (16.56 mg/g) in accession EC182945 (Fig. S2G). For Gli content, all the accessions belonged to low category with the range of 26.35–44.94 mg/g (Fig. S2H). In contrast, all accessions were grouped in high Glu content class with the mean of 31.00 mg/g and ranged from 24.47–39.56 mg/g (Fig. S2I). Gli and Glu are collectively called gluten which shares 85% of total protein (Fisher & Halton, 1936) and is responsible for visco-elastic property to the wheat dough. By summing up these two fractions of protein, we got a rough estimate of gluten content. With an average of 64.97 mg/g proportion of gluten among accessions, it varied from 55.33–75.06 mg/g (Fig. S2J). Significant positive correlation of gluten with Gli (r = +0.73**) and Glu (r = +0.60**) depicted collective contribution of these components in gluten estimation. Regarding lysine content, T. sphaerococcum wheat had an average of 0.15% of sample Lys with a range of 0.04% (EC187172) to 0.29% (EC10492) (Fig. S2K). We also observed non-significant and negative correlation (r = −0.03) between lysine and protein whereas significant negative (−0.35*) association was found with gluten (Table 4).

Table 5 Categorization of T. Sphaerococcum accessions in low, medium, and high category for protein fractions.

Accessions	Alb (mg/g)	Glo (mg/g)	Gli (mg/g)	Glu (mg/g)	
IC384530	L	M	L	H	
IC397363	L	L	L	H	
IC533826	L	L	L	H	
IC53387	L	M	L	H	
IC534021	L	M	L	H	
IC534522	M	L	L	H	
IC212160	L	M	L	H	
IC534882	L	L	L	H	
IC420038	L	L	L	H	
EC313761	L	M	L	H	
EC541164	L	L	L	H	
EC576654	L	M	L	H	
EC613055	L	L	L	H	
EC613057	L	L	L	H	
EC187172	L	M	L	H	
EC187182	L	M	L	H	
EC187181	L	M	L	H	
EC180062	L	M	L	H	
EC187183	L	L	L	H	
EC187167	L	L	L	H	
EC182959	L	L	L	H	
EC182945	L	M	L	H	
EC10494	L	M	L	H	
EC10492	L	M	L	H	
EC182958	L	L	L	H	
EC180041	L	L	L	H	
EC180061	L	M	L	H	
EC180067	L	L	L	H	
EC182947	L	M	L	H	
EC180066	L	L	L	H	
EC182956	L	M	L	H	
EC180063	L	M	L	H	
EC187176	L	L	L	H	
IC634028	L	M	L	H	
Notes:

Alb-Albumin: Low = <40 mg/g.wt., Medium = 40.1–50 mg/g.wt., high = 50.1–55 mg/g.wt.

Glo-Globulin: Low = <13 mg/g.wt., Medium = 13.1–17 mg/g.wt., High = 17.1–21 mg/g.wt.

Gli-Gliadins: Low = <55 mg/g.wt., Medium = 55.1–80 mg/g.wt., High = 80.1–93 mg/g.wt.

Glu-Glutenin: Low = <12 mg/g.wt., Medium = 12.1–16 mg/g.wt., High = 16.1–20 mg/g.wt.

Fe2+ and Zn2+ content

Among micronutrients, T. sphaerococcum accessions had a mean of 65.11 ppm Fe2+ content with a range of 42.72 ppm (EC180066) to 90.72 ppm (IC384530) (Figs. S2l and 3), whereas slightly narrow range was observed for Zn2+ content i.e., from 11.45 ppm to 25.70 ppm in accession EC182956 and EC10492 respectively (Fig. S2M). Fe2+ content showed a highly significant positive correlation with major protein fraction i.e., Gli (r = +0.48**) and Glu (r = +0.52**) (Table 4). Likewise, Zn2+ content was positively (+0.55**) associated with HI. Zn2+ concentration showed a non-significant weak correlation with TGW (0.22).

Enzymatic antioxidant activity

Among four studied antioxidants, POX activity ranged from 0.06–0.60 units/ml with an average of 0.22 units/ml (Figs. S2N and 4). The accession “EC541164” possessed the highest POX activity. POX is considered important from the quality point of view due to its involvement in the reduction of dough adhesiveness. The CAT, GR and SOD activity varied from 0.02–0.60 unit/ml, 0.11–2.26 unit/ml, and 0.14–0.97 unit/ml respectively (Figs. S2O–S2Q). Correlation analysis results indicated that only GR and POX showed a positive correlation with HI. Whereas negative relation was observed between POX and Pro, GR and HW, CAT with Sed and Alb, and SOD with Zn2+content.

Cluster analysis

Thirty-four T. sphaerococcum accessions were subjected to cluster analysis by employing an unweighted pair group method with arithmetic average (UPGMA) and Euclidean distance measure. Euclidean distance matrix depicts the extent of genetic dissimilarity among accessions. The maximum distance of 86.22 was observed between EC613055 which belonged to cluster B and IC384530 which is clustered independently in cluster D from the rest of the accessions. After that EC187172 and IC384530 with 75.97 euclidean distances were grouped in clusters B and D respectively. EC187183 and IC420038 were most similar due to the lowest euclidean distance of 6.57 and both of them belonged to cluster C. Following this, pair of EC613055/EC576654 clustered in group B with a distance of 7.02. The clustering pattern of Indian dwarf wheat accessions is based on quality data and is presented in Fig. 5 and Table 6. Based on quality parameters, at a Euclidean distance of 30, 34 accessions of Indian dwarf wheat were grouped into four main clusters A, B, C, and D. Further main cluster A, B, and C is divided into two sub-clusters each whereas only one accession was grouped in main cluster D. The distribution pattern revealed the maximum number of genotypes i.e., 17 in cluster C followed by cluster B having 10 genotypes. The cophenetic correlation coefficient was estimated as 0.86 for checking the agreement between phenograms and dissimilarity matrices for the accuracy of the dendrogram. The mean performance of different clusters for the studied quality traits was shown in Table 6. Minimum mean Sed (29.45 ml) was observed among accessions of cluster B. In addition to this, cluster B was also characterized by low Gli (31.40 mg/g), Glu (30.04 mg/g), gluten content (61.44 mg/g), and high HI (90.24). Cluster B also showed a high mean value for POX (0.26 unit/ml) and GR (0.78 unit/ml) activity and hence could be targeted for wheat nutritional quality improvement. For flour yield improvement accessions clustered in cluster C could be targeted due to the maximum mean HW (81.82 kg/hl). Accessions grouped in cluster C also showed high mean values Alb (24.96 mg/g) and Lys (0.19%), whereas cluster D was characterized by high Glo (14.17 mg/g). In addition to high Glo accession of cluster D also showed high Sed (32.00 ml), Gli (41.78 mg/g), gluten (72.68 mg/g), Fe2+ (90.71 ppm), and SOD activity (0.63 unit/ml). Cluster A demonstrated high Glu (33.26 mg/g), Zn2+ (22.56 ppm) content, CAT (0.16 unit/ml), and GR (0.78 unit/ml) activity. To understand and confirm the dynamics of the sphaerococcum accessions, we have further carried out principal coordinate analysis (PCoA). Based on the pairwise Euclidean distance matrix, the clear distinction could be visualized among these accessions, which was in concordance with the results of the biplot (Fig. 6).

Figure 5 Clustering pattern of 34 T. sphaerococcum accessions based on eighteen quality traits.

Table 6 Four clusters constituting of different genotypes along with mean values of clusters for 17 quality traits.

Cluster	A	B	C	D	
No. of genotypes		6	10	17	1	
Genotypes	EC187176, EC182945, EC182958, EC10494, EC187182, EC10492	EC182959, EC182956, EC187172, EC180061, EC613055, EC576654, EC187181, EC187167, EC182947, EC180066	EC180063, EC313761, IC397363, IC533826, IC534021, IC420038, EC187183, IC53387, EC541164, IC534882, EC613057, EC180062, EC180041, IC212160, IC634028, IC534522, EC180067	IC384530	
Mean values of clusters for seventeen quality traits	TGW(g)	35.30	31.84	32.70	31.11	
HW (kg/hl)	79.58	81.15	81.82	82.50	
Sed (ml)	31.42	29.45	30.47	32.00	
HI	50.92	90.24	88.45	81.07	
Pro (%)	18.38	17.20	17.51	18.90	
Alb (mg/g)	24.12	22.39	24.96	20.13	
Glo (mg/g)	14.10	13.29	13.07	14.17	
Gli (mg/g)	35.02	31.40	34.65	41.78	
Glu (mg/g)	33.26	30.04	30.78	30.90	
Gluten (mg/g)	68.28	61.44	65.42	72.68	
Lys (% of sample)	00.10	0.13	0.19	0.16	
Fe2+ (ppm)	65.43	65.10	65.40	90.71	
Zn2+ (ppm)	22.56	17.22	18.95	16.25	
POX (unit/ml)	0.11	0.26	0.23	0.12	
CAT (unit/ml)	0.16	0.10	0.10	0.15	
GR (unit/ml)	0.78	0.78	0.66	0.11	
SOD (unit/ml)	0.22	0.36	0.28	0.63	
					
Note:

TGW, Thousand grain weight; HW, Hectolitre weight; Sed, Sedimentation value; HI, Hardness index; Pro, Protein content; Alb, Albumin content; Glo, Globulin content; Gli, Gliadin content; Glu, Glutenin content; Lys, Lysine content; POX, Peroxidase; CAT, Catalase; GR, Glutathione reductase; SOD, Superoxide dismutase.

Figure 6 Biplot representing distribution pattern of 34 T. sphaerococcum accessions.

Validation of high grain protein accessions

Eight sphaerococcum accessions with high grain protein content (GPC) were validated under multi-location and multi-year trials by growing consecutively at NBPGR, Pusa Farm and NBPGR, Issapur Farm, New Delhi during the year 2019–2020, 2020–2021, and 2021–2022. Their comparative performance along with the overall mean is presented in Fig. 7. Top performing accessions with more than 15% GPC were confirmed as EC182958 (17.16%), EC187176, and EC182945 (16.16%), EC613057 (15.79%), IC634028 (15.72%) and IC533826 (15.01%) which can be used for introgression purpose and bread wheat quality improvement.

Figure 7 Depiction of mean performance of 8 high grain protein accessions over the locations and years through line diagram.

Discussion

Wild relatives and ancient wheat serve as the most important genetic resources for numerous valuable traits and possess wider variation for quality traits (Zeibig, Killian & Frei, 2021). In comparison to bread wheat (T. aestivum), emmer wheat (T. dicoccoides) is reported to possess more lysine and isoleucine content (Nevo & Beiles, 1992). Similarly, as appose to cultivated wheat, and wild relatives; primitive cultivars are considered better resources for zinc and iron fortifications (Cakmak, 2008; Velu et al., 2011; Xu et al., 2011). Extensive characterization of genetic resources is a preliminary step for designing any quality breeding program. The ancient wheat, T. sphaerococcum, endemic to north-western India and southern Pakistan was withdrawn from cultivation during the early 20th century due to the introduction of the modern wheat cultivar as the result of the green revolution (Mori et al., 2013). Indian dwarf wheat is the least exploited hexaploid species for quality assessment. Here we explored T. sphaerococcum germplasm for their nutritional quality traits to enhance their potential utilization for bread wheat improvement.

In our study, a nearly similar range of HW was observed in T. sphaerococcum as reported by Mohan & Gupta (2013) in bread wheat cultivars. Higher HW means better milling quality in terms of flour yield (Martin et al., 2021) and thus accessions coming under the category of higher HW may result in better flour yield. As, HW is affected by grain shape, size, and density, therefore the possible reason for the higher value is their round grain shape. Sed recorded among these accessions was much lower than Indian bread wheat varieties (35–60 ml, (Mohan & Gupta, 2013)) and (33–52 ml, (Panghal, Chhikara & Khatkar, 2017)). Whereas, regarding the hardness index, most of the studied accessions belonged to the hard category. However, a relatively wide range (29–94) for HI in Indian bread wheat varieties was reported by Mohan & Gupta (2013). It has been observed that wheat with high protein tends to be hard whereas soft grain with low protein.

Wheat constitutes approximately 80% of the total cereal intake, and it contributes 50% of total energy and 60% of the total protein intake. Based on our findings, sphaerococcum wheat grain accumulated more protein in contrast to the narrow range (9.40–13.50) of protein in Indian bread wheat varieties reported by Mohan & Gupta (2013). We have also confirmed the high GPC accessions by growing in different environments. However, Ciaffi et al. (1992) found a large variation in T. dicoccoides, ranging from 16.00–27.00%. Similarly, in related species of wheat, mean Pro was reported as 16.67% (Jiang et al., 2008) confirming the finding that wild and primitive wheat have a wider range of variation for grain protein than cultivated ones. The Alb and Glo contribute 15–20% share of wheat flour protein. These non-prolamin protein fractions are considered nutritionally better due to a higher proportion of lysine and methionine as compared to other proteins in wheat (Lasztity, 1986). In addition to this, these proteins are also considered important for germinating and protecting embryos from insects and pathogens (DuPont & Altenbach, 2003). In our study, we have identified one accession for moderate Alb and eighteen for moderate Glo. These accessions were considered good from the nutritional point of view and could be utilized in wheat quality improvement. The Gli and Glu ratio in the present study ranged from 0.80 to 1.50 which is lower than as observed in the case of common wheat (1.60–3.80) (Geisslitz et al., 2019). It has been proven that a low Gli:Glu ratio is linked with good baking quality (Thanhaeuser, Wieser & Koehler, 2015). Therefore, these accessions are also considered good for baking purposes. Cereal-based diet and food items are poor concerning essential amino acids such as lysine and methionine (Smriga et al., 2004). Inadequate intake of Lys results in a syndrome called ‘protein energy malnutrition’ which is suspected to be more prevalent in developing countries. Manipulation of cereal proteins and amino acid profiles will cause a significant impact on the life of hundreds of millions of poor people. Despite extensive research aimed at the identification of Lys-rich wheat accessions; the high Lys mutants have not been discovered yet. Accessions with higher Lys estimates (EC10492, EC187176, EC182945, IC534522, and EC313761) found in our study could be utilized in wheat quality improvement.

The iron content in T. sphaerococcum accession was much higher than those obtained by Zhao et al. (2009), Mohan & Gupta (2013), Goel et al. (2018) and Khokhar et al. (2020). However, variation for Zn content is less as appose to previous findings (Goel et al., 2018; Khokhar et al., 2020) whereas in agreement with the findings of Zhao et al. (2009). It has been observed that grains with high Pro accumulate more micronutrients, which may be due to the co-localization of Pro, Zn2+, and Fe2+ in the embryo and the aleurone layer of the grain (Khokhar et al., 2020). Zn2+ concentration showed a non-significant weak correlation with TGW (0.22) which is in close agreement with the findings of Khokhar et al. (2020). A significant positive association (+0.43*) between Fe2+ and Zn2+ contents indicated the possibility of simultaneous improvement of both micronutrients (Goel et al., 2018). It was also stipulated that there is the involvement of a common mechanism for the accumulation of both micronutrients (Hernandez-Espinosa et al., 2020). Likewise, a significant positive association between Fe2+ and Zn2+ was observed by various researchers (Velu et al., 2017; Hernandez-Espinosa et al., 2020; Khokhar et al., 2020). Further, the probable reason for the association between Fe2+ and Zn2+ content was found to be the co-localization of quantitative trait loci present at chromosome 3B governing these traits (Velu et al., 2017). Due to the lack of ample genetic variation in cross-compatible species, the traditional breeding for micronutrient enhancement was non-exploitable. Henceforth hexaploid species T. sphaerococcum could be exploited in this direction. For antioxidants, maximum CAT, GR, and SOD activity were recorded in the case of accessions EC18294, EC180063, and EC182956 respectively. CAT was reported to be involved in oxidative reaction during bread making and its highest activity was recorded during the kernel development stage and was responsible for the detoxification of H2O2 (Kruger & Leberg, 1974). Likewise, SOD protects the oxidation of cellular components through reactive oxygen species (Mughal et al., 2020). Hence accessions showing high activity for antioxidants could be targeted for wheat quality improvement in terms of nutrition. GR may control the ratio of GSSG (oxidized glutathione)/GSH (reduced glutathione) in wheat flour and play a role in determining the physical properties of wheat flour doughs (Every et al., 2006).

Based on association analysis, there were non-significant positive estimates of the correlation coefficient between Sed and Pro; however, it was significantly correlated as per the findings of Drikvand et al. (2013). Further, a significant positive correlation was observed between Pro and HI which is supported by the finding of Drikvand et al. (2013). HI and TGW were negatively associated at 0.05% level of significance, likewise non-significant but negative correlation estimates were observed by Abdipour et al. (2016). Also, it has been proven that Lys is negatively correlated with Pro as reported by Rharrabti et al. (2001).

The clustering pattern based on quality parameters revealed 17 accessions in cluster C followed by cluster B having 10 genotypes with cophenetic correlation coefficient value as 0.86 greater than the acceptable level of 0.85 (Stuessy, 2009). Minimum mean Sed was observed among accessions of cluster B. In addition to this, cluster B was also characterized by low Gli, Glu, gluten content, and high HI. Cluster B also showed a high mean value for POX and GR activity, hence, could be targeted for wheat nutritional quality improvement. For flour yield improvement, accessions clustered in cluster C could be targeted due to the maximum mean HW. Accessions grouped in cluster C also showed a high mean value of Alb and Lys, whereas cluster D was characterized by high Glo. As Alb and Glo protein is reported nutritionally better due to a higher proportion of lysine and methionine (Lasztity, 1986) therefore these accessions could be utilized in common wheat improvement for essential amino acids. In addition to high Glo, accession of cluster D also showed high Sed, Gli, gluten, Fe2+, and SOD activity. Also, accession of cluster D could be targeted for enhancement of protein content in common wheat as they are characterized by high protein content. Cluster A demonstrated high Glu, Zn2+ content, CAT, and GR activity henceforth could be exploited in the improvement of respective traits.

Future implications

Based on the comprehensive estimation of grain quality parameters, our study revealed that the primitive wheat landrace, Indian dwarf wheat T. sphaerococcum is a rich source of grain protein content, lysine content, iron and zinc content, and antioxidant activity. It also has low gluten content which is a major dietary requirement based on consumer preference. Therefore exploration, conservation, and use of collections belonging to this species may be promoted to gain maximum benefit for the well-being of mankind. Among the studied accessions, EC10492, and EC313761 (from USSR) were found superior for a maximum of seven traits followed by EC180063 and EC182958 for five traits. We also validated high GPC accessions such as EC182958 (from Canada), EC187176, EC182945, EC613057 (from France), IC634028 (Kathod Gehun from Tapi district of Gujarat), and IC533826 with more than 15% grain protein content.

Conclusions

Wild relatives, ancient species, and landraces are rich valuable genetic resources for abiotic and biotic stress tolerance, and nutritional qualities and play a major role in wheat improvement. In the present study, we analyzed 34 T. sphaerococcum, Indian dwarf wheat accessions conserved in the national genebank of India for seventeen-grain quality parameters. We found sufficient variation among accessions for all studied quality traits and accessions with superior quality traits were identified. Among these accessions, EC10492, and EC313761 were found superior for a maximum of seven traits followed by EC180063 and EC182958 for five traits. We also validated high GPC accessions and EC182958 (17.16%), EC187176 and EC182945 (16.16%), EC613057 (15.79%), IC634028 (15.72%) and IC533826 (15.01%) were confirmed with more than 15% GPC. The superior accessions identified in the present study could be the potential genetic resource for a breeder to improve the wheat’s nutritional quality. The sphaerococcum wheat should be promoted for cultivation on the farmers’ field considering its richness in grain nutrients and end-use quality with low gluten, high protein, high iron, and zinc content, and high antioxidant activity. Further, the germplasm explorer should prioritize the collection of this species from unexplored areas to harness its nutritional richness for alleviating nutritional imbalance and improving the health of mankind. The results of the present study showed the possibility of utilization of Indian dwarf wheat in wheat quality breeding aiming for nutritional security under the United Nation’s sustainable development goals (SDG).

Supplemental Information

Supplemental Information 1 Raw Data.

Click here for additional data file.

Supplemental Information 2 Flow of work done for grain quality analysis.

Click here for additional data file.

Supplemental Information 3 Histogram depicting variations in different grain quality traits studied.

a) Hectolitre weight (HW), b) Thousand grain weight (TGW) c) Sedimentation value (Sed), d) Hardness index (HI), e) Protein content (Pro), f) Albumin content (Alb), g) Globulin content

Click here for additional data file.

The authors wish to acknowledge the Director of the ICAR-National Bureau of Plant Genetic Resources, New Delhi for extending infrastructure facility for carrying out the research work. The authors also wish to thank the explorers for collecting the germplasm and farmers for maintaining valuable genetic resources.

Additional Information and Declarations

Competing Interests

Author Contributions

Data Availability

Sapna Langyan is an Academic Editor for PeerJ.

Sneha Adhikari conceived and designed the experiments, performed the experiments, analyzed the data, prepared figures and/or tables, authored or reviewed drafts of the article, and approved the final draft.

Jyoti Kumari conceived and designed the experiments, performed the experiments, analyzed the data, prepared figures and/or tables, authored or reviewed drafts of the article, and approved the final draft.

Rakesh Bhardwaj performed the experiments, prepared figures and/or tables, authored or reviewed drafts of the article, and approved the final draft.

Sherry Jacob performed the experiments, authored or reviewed drafts of the article, and approved the final draft.

Sapna Langyan performed the experiments, authored or reviewed drafts of the article, and approved the final draft.

Shivani Sharma performed the experiments, prepared figures and/or tables, and approved the final draft.

Anju M. Singh performed the experiments, authored or reviewed drafts of the article, and approved the final draft.

Ashok Kumar conceived and designed the experiments, authored or reviewed drafts of the article, and approved the final draft.

The following information was supplied regarding data availability:

The raw data is available in the Supplemental File.

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
