# Peer review of "Unlocking the potential of ancient hexaploid Indian dwarf wheat, Tritium sphaerococcum for grain quality improvement"

_PeerJ, doi:10.7717/peerj.15334_

## Round 0.1 · original submission · Major Revisions

Both of the reviewers found that the manuscript has scientific merit but need significant improvement. One of the reviewers has raised questions about the data presentation and statistical analysis. The statistics must be improved with proper mean seperation methods. The manuscript has some language issues and these should be addressed. The authors are requested to submit the revised manuscript with pointwise responses.

Reviewer 1 ·

Basic reporting

The manuscript, "Unlocking the potential of ancient hexaploid Indian dwarf wheat, Tritium sphaerococcum for grain quality improvement," is very interesting and contains some important findings. But the authors need to improve the presentation of the data, especially the figures. The English language and grammar is satisfactory.

Experimental design

The authors mentioned that the grain harvest was obtained from the field trial grown during Rabi 2019–20. But they did not mention any cultivation process, fertilization, experimental design, the initial soil fertility status, as well as the climatic condition of the experimental site, but the grain quality is significantly affected by these factors. They must surely need to include these. As the quality analysis was performed in a controlled laboratory condition, the data needed to be analysed with CRD design with proper post-hoc analysis along with the Sed values.

Validity of the findings

No comment

Additional comments

1. The correlation table needs to be present in the main text.
2. In the discussion section, a section with the future implications of the study needs to be included.
3. Table 3: List the actual values of Alb, Glo, Gli, and Glu in the table and categorise them as low, medium, or high. Add extra columns for that.
4. It would be better to add a biplot along with Table 4 for better understanding.
5. Figure 1 may be going to the supplementary file or the graphical abstract (if the journal permits).
6. Figure 2 needs to be re-drawn. It is not visible. It is better to make different figures with error bars or letter differences (it is better to perform the DMRT/LSD Test) and add axis legends for both the X and Y axis.
7. Figure 3 is also not clearly visible. It is better to make a clear figure with error bars or letter differences (better to perform the DMRT/LSD Test) and add axis legends for both the X and Y axis.
8. Figure 4: It is good to provide a cluster diagram. But if authors recast the cluster diagram into a circular dendrogram, it will be very attractive. Authors may take the help of LeOra Polo Plus Software or other tools.

Reviewer 2 ·

Basic reporting

No comment

Experimental design

No comment

Validity of the findings

No comment

Annotated reviews are not available for download in order to protect the identity of reviewers who chose to remain anonymous.

---

## Round 0.2 · Minor Revisions

Based on the comments received by the reviewers, the revised manuscript is acceptable for publication but needs considerable language editing. An annotated manuscript with some sugggestions is attached herewith. Please check the grammar and language meticulously before submission.

Reviewer 1 ·

Basic reporting

Clear and unambiguous, professional English used throughout.

The paper is now in good condition.

Experimental design

Within the scope of the journal and well structured.

Validity of the findings

No comments.

Additional comments

The authors have made a substantial revision to the manuscript by addressing all comments. Hence the manuscript may be accepted for publication.

---

## Round 0.3 · accepted · Accept

The original Academic Editor is unavailable so I am sending the decision in my capacity as Section Editor.

The present version of the manuscript, entitled "Unlocking the potential of ancient hexaploid Indian dwarf wheat, Tritium sphaerococcum for grain quality improvement" has been improved significantly and the manuscript is now ready for acceptance. The manuscript reads well and highlights the breadth of traits exhibited within the sphaerococcum species. As a potential donor for moving desired traits into new bread wheat germplasm the present manuscript appears to highlight the expectations for plant breeders. The material would be of interest for the readership contemplating such an endeavor. The manuscript appears suitable for publication and is recommended for final approval. Congratulations.